# Additive Manufacturing of Complex Components through 3D Plasma Metal Deposition—A Simulative Approach

**Khaled Alaluss [1],* and Peter Mayr [2]**

[1]  Steinbeis Innovation Center Intelligent Functional Materials, Welding and Joining Techniques, Implementation, Manfred-von-Ardenne-Ring 20, 01099 Dresden, Germany

[2]  Chair of Welding Engineering, Chemnitz University of Technology, Reichenhainer Strasse 70, 09126 Chemnitz, Germany; peter.mayr@mb.tu-chemnitz.de

*  Correspondence: khaled.alaluss@stw.de; Tel.: +49-351-8925-422

**Abstract:** This study examines simulative experimental investigations on the additive manufacturing of complex component geometries using 3D plasma metal deposition (3DPMD). Here, complex contour surfaces for a cross-rolling tool were produced from weld metals in multilayer technology through 3DPMD. As a consequence of the special features of 3DPMD with large-weld metal volumes, greatly differing properties between base material/deposited material and asymmetrical heat input, the resulting shrinkage, deformation and residual stresses are particularly critical. These lead to dimensional and form deviations as well as the formation of cracks, which has a negative influence on the quality of the plasma deposition-welded component structures. By means of the thermo-elastic-plastic simulation model, the temperature field distribution, deformation, and residual stresses occurring during additive 3DPMD of tool contours were predicted and analyzed. The temperature field distribution and its gradients were determined using the ellipsoid heat-source model for the 3DPMD process. On this basis, a coupled thermo-elastic-plastic structural–mechanical analysis was performed. Accordingly, the results achieved were used for the production of almost-net-shaped tool contour surfaces with predefined layer properties. The acquired simulation results of the temperature fields, deformation, and residual stress condition show good alignment with the experimental results.

**Keywords:** additive manufacturing; 3DPMD; filler materials; tool model; FE model; simulation; warpage and residual stresses; temperature fields; heat source model

## 1. Introduction

The sophisticated additive manufacturing technology for processing metallic materials by means of 3D plasma metal deposition (3DPMD) from pure large-weld metal volumes is establishing itself as a new manufacturing process. The specific advantages of 3D additive metal deposition are its material and geometric flexibility as well as reduction of scrap materials and, thus, the reduction of production costs. In addition, complex component geometries that are difficult or impossible to manufacture using conventional manufacturing processes can be produced using this technology. In particular, the enormous geometric freedom associated with this process technology enables the production of complex component geometries such as functionally graded components [1–4].

A lot of processes are used in wire arc additive manufacturing (WAAM) for e.g., metal inert gas (MIG), (cold metal transfer) CMT, tungsten inert gas (TIG), etc. The CMT have relatively low thermal input and almost none sputter. Therefore, it is supposed to produce a deposited metal with excellent quality. This is true for steels and aluminum but for titanium, the surface roughness of the

deposited metals is relatively high, due to the arc wandering. One of the limitations of the WAAM process is the generating of residual stresses at the deposited layer(s), which leads to distortions. Furthermore, geometrical dimension accuracy limits the wide use of this technology in the industry. Thus, in order to optimize the 3DPMD, many techniques have been applied; for example, *Symmetrical building*: this approach is useful in reducing the stresses due to stresses balancing. One deposited side will generate stresses; these will balance with other stresses from its symmetrical side. *Back to back building*: in this approach, the components will be built on either side of the same substrate, which, in this case, is sacrificial. In this approach, the management of heat can be improved. *Optimizing part orientation*: this approach depends on producing a component using a shorter possible path for the layer. *High pressure interpass rolling:* in this technique, a rolling process was performed after each layer. This process leads to the migration of residual stresses [5].

Venturini [6] showed that deposition strategies have a clear influence on deposition efficiency, which is defined as the ratio between the final part volume (after finishing operation) and the total deposited volume. In addition, the deposition strategy affects the generated stresses [7–9]. Martina et al. [8] applied rolling within the deposition process to reduce grain growth as well as to reduce the effect of the residual stresses [8]. Note that metallurgical aspects are associated with the additive manufacturing (AM) process; thus, the aluminum additive manufacture process suffers from the porosities as well as nonuniform grain growth due to the relatively high heat input. These porosities may have a negative influence on yield fatigue characteristics for aluminum. The porosities within 50–100 μm in mean diameter act as crack initiation sites [5,10]. In addition, the WAAM process for aluminum suffered from hydrogen existence in the welding pool through the welding material is produced. In WAAM process, this problem is more difficult to control, due to the filler metal which is the main source of hydrogen. Moreover, a large amount of filler metal is fed into the molten pool [10]. Gu et al. [10] succeed to eliminate the porosities with a diameter larger than 5 μm in aluminum alloys samples (2319, 5087) that produced using WAAM process. The elimination techniques were cold work (rolling) and post-rolling heat treatment. The rolling process was performed in each deposited layer at a load of 45 kN.

In the plasma metal deposition process, a wire or powder as filler metals can be used for the production of metallic components; further, 3D contours can be generated by employing multilayer deposition technology, as 3DPMD permits the layer-by-layer production of metallic components based on a virtual CAD (computer aided design) component model. Typical layer heights are from 1.0 to 5.0 mm in single-pass deposition welds. By mixing several powders in an arc, the local properties of the added layer can be adapted to the defined service loads, locally and partially. With this technology, a minimum thermal load on the deposit and base materials is achieved; in addition, a reliable and reproducible layer quality is guaranteed [11–13]. In [11–13] the process-specific advantages of the 3D additive manufacturing technology by means of the aforementioned fusion welding processes are presented and discussed. The geometric flexibility of this process technology, which is particularly flexible in the production of complex component geometries, should be emphasized. One exception is the use of the arc-based microplasma process (μ-PTA) in additive manufacturing. The welding performance and the feed rate of the filler material are reduced so that low build-up rates of just a few grams per minute can also be achieved for the production of thin-walled components with high final contour accuracy [13].

The final microstructure is affected by the heating regime of the AM process, due to excitation of the alloying elements. The AM process was applied to produce different types of base metals. Using weldable alloys is easier than using non-weldable alloys, as non-weldable alloys generate intermetallic phases in addition to other metallurgical problems. Use of finite element (FE) modeling to calculate the residual stresses and their relation to the AM parameters, deposition strategy, the effect of the materials type, and the thermal conditions seems to benefit advances in optimizing the AM process [9,14,15].

In the investigations, the first segment of a cross-rolling tool from the metal processing was included. Its contoured surfaces were produced using 3DPMD. The produced tool was used as an application component. The cross-rolling tool contour surfaces were produced using 3DPMD with predefined layer

properties. In practice, this tool is subject to complex thermomechanical loads/stresses. High wear, corrosion, and temperature loads are generated at the tool surfaces. In addition, pressure-shear stress, thermally induced stresses, and wear effects occur due to abrasive effects on the tool surfaces, which are subjected to high thermal loads due to high rolling temperature up to 1200 °C [2,13]. Due to the many advantages described above for specific processes, the tool is manufactured using additive 3DPMD. The aim of this investigation was to increase tool life by improving its thermomechanical properties as well as to predetermine the temperature field distribution and to optimize the 3DPMD by means of the thermomechanical simulation model. Consequently, the resulting deformation and residual stresses can be minimized or reduced. The effects of modifications in thermal deposition technology such as optimization of process parameters/cooling conditions, base body preheating, and fixed clamping such as application of layers with good suitability properties were determined, demonstrated, and discussed using the FE model with ANSYS®. Based on the knowledge gained, an additive 3DPMD strategy for the production of a cross-rolling tool with functional surface properties, e.g., wear and temperature resistance by the effect of the workpiece, was developed and implemented in practice.

## 2. Approach–Development of Innovative Three-Dimensional Plasma Metal Deposition (3DPMD) Technology

### 2.1. Experimental Work

#### 2.1.1. Materials

For the development of error-free 3DPMD technology for the production of contour surfaces for the selected application component (i.e., cross-rolling tool), the base material 1.2344 hot work tool steel (X40CrMoV5-1) was used. This material is mainly used for the production of cross-rolling tools by means of subtractive technologies. In order to improve the tool life in practical use, heat-resistant cladding materials were used to produce contours. The selected filler materials have good abrasive wear resistance, oxidation resistance, scale resistance, and high-temperature resistance. Table 1 shows the types and chemical compositions of the applied materials. Hard powder materials with a grain size of 50–150 μm are made of iron-based alloys, PS Fe-hard D, and EuTroloy 16604 alloys were used in this investigation. The hardness values of these alloys are 65 and 46 HRC (rockwell-hardness), respectively. These alloys have high wear resistance under abrasive and fatigue load as well as high wear resistance for a combination of abrasion and fatigue. Also, they have a relatively high hardness at high temperatures. The iron-based powders (PS Fe-hard D and EuTroloy 16604) were mixed with each other in different fractions to achieve a milder and high hardness at high temperatures for the layers to be deposited onto the tool contour surfaces. The cobalt-based alloy Stellite 12 HC was used as reference material in the investigations. These alloys have high abrasion resistance under pressure and impact stress at high temperatures. The powder grain size for this alloy is 63–150 μm and the hardness is 52 HRC. In order to compensate for the large differences between the material properties of the base and hard coating materials, a sufficiently tenacious and heat-resistant layer of nickel-based alloy Ni 625 with a grain size of 63–180 μm was used for the production of the tool contour surfaces, which had a maximum hardness of 23 HRC.

**Table 1.** Types and chemical composition of the used materials in the investigations [16,17].

| Materials | Chemical Composition (wt.%) | | | | | | | | | |
|---|---|---|---|---|---|---|---|---|---|---|
| | C | Cr | Co | Mn | Mo | Ni | Fe | Si | V | W |
| X40CrMoV5-1 | 0.40 | 5.2 | - | 0.4 | 1.3 | - | bal. | 1.0 | 1.0 | - |
| PS Fe-hard D | 1.0 | 4.0 | - | - | 5.0 | - | bal. | - | 2.10 | 6.20 |
| EuTroloy 16604 | 0.20 | 15 | 15 | - | 2.5 | - | bal. | - | - | - |
| Stellite 12 HC | 1.90 | 32 | bal. | - | - | <3.0 | <3.0 | 1.0 | - | 9.5 |
| Ni 625 | 0.03 | 21 | - | - | 8.60 | bal. | - | 0.50 | 3.40 | - |

### 2.1.2. Process Parameter Determination for 3DPMD of a Layered System

The aim of these investigations was to develop a suitable multilayer construction system with better thermomechanical properties for this tool. For the determination of optimal process parameters in 3DPMD, a $2^3$ statistical method was used (i.e., two-level method). Thus, for three variable parameters, each with a high and low value, only $2^3 = 8$ tests are needed to determine their influence on the target size. The welding tests were carried out under the same welding conditions. The process parameters such as current, voltage, linear and weaving speed, plasma conveying, and shielding gas as well as powder quantity were varied and characterized in accordance with the layer character during the experimental investigations.

This results in quality considerations taking into account the main target: layer build-up; hardness of the depositions; degree of mixing; deposition layer geometry and structure. The result was a homogeneous, fine-grained microstructure, high hardness, and, at the same time, crack-free and uniform deposition, which requires little reworking of the component functional surfaces. Multilayer depositions in linear and weaving PMD were realized. Care was taken to reduce the heat input and the energy per unit length in multilayer thermal deposition from layer to layer. Figure 1 shows examples of 3D plasma-deposited samples.

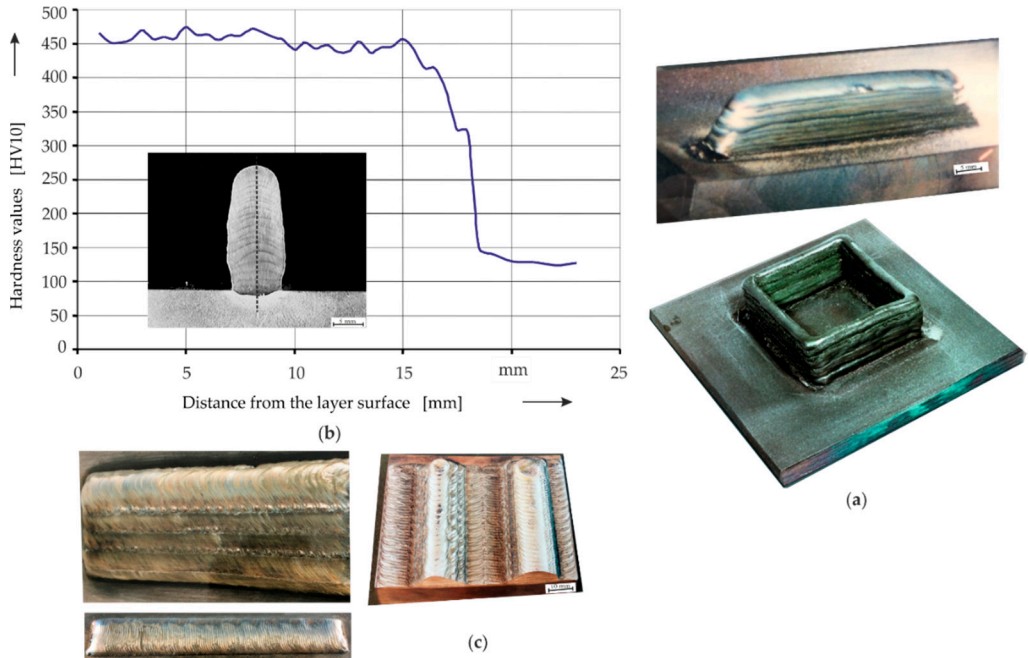

**Figure 1.** Three-dimensional plasma metal deposition (3DPMD) samples: (**a**) Linear PMD contours, (**b**) hardness profiles for linear PMD and (**c**) weaving PMD contours.

### 2.1.3. Characterization of the Plasma Deposition-Welded Layer System

Metallographic investigations were carried out after preparing the specimens to characterize the microstructure of the deposited layers. Qualitative and quantitative evaluations were made with consideration of the microstructure transition from the base body to the hard surface passing through the deposit welds. Hardness characteristics were determined perpendicular to the deposited layers on the produced macro-sections using the Vickers method according to DIN 50133 [18] with the macro-hardness of HV5, which were converted into Rockwell values for evaluation. This allowed us to state the mechanical-technological properties at the surface and for subsurface areas.

To guarantee realistic and load-adapted investigations, an abrasive sliding wear test was performed under a thermal load according to [19]. By adding an intermediate material, such as quartz sand, it is possible to simulate a three-body sliding wear test. The abrasive effect is caused by rotation of the

Hardox disc. Friction speeds between 10 mm·s$^{-1}$ and 1500 mm·s$^{-1}$ can thus be achieved. The contact pressure of the test specimens on the disc can be adjusted variably up to 100 MPa. Another advantage is the possibility to test at different temperatures up to 800 °C. Thus, the field of application of the applied parts (i.e., tool) was realistically simulated. This enables the tool to be subjected under high-abrasive as well as high-thermal loads.

For verification of the FE model, measurements were performed on the deposited welded specimens with regard to the temperature field distributions, weld deformations, and residual stresses. The workpiece surface temperatures were determined for two- and four-layer welds using attached Pt/Pt-Rh thermocouples. Validation of the FE calculations performed regarding the residual stresses and deformations that occur in the deposit welded specimens was realized by applying the measurement coordinate system for the deformation measurement and the hole-drilling method [20] for the residual stress measurement. The drill-hole method used can be used to drill out blind holes up to a depth of 1.8 mm. From the investigations performed, drilling deeper than 1.0 mm in the hard layer gives no more reliable information related to strain changing. The strain to be determined depends on the height of the residual stresses in the respective depth increment and the geometric conditions (drill diameter, distance of the strain gages to the drill hole edge, strain gage geometry, etc.). The stresses reached its maximum value at the surface of the deposited layer. The inter pass temperature (heat treatment process) reduces the stresses in deeper distances. As the number of layers' increases, the stress values on the surface of the last weld layer increase rapidly due to the faster surface cooling of the last weld layer. This measurement was carried out on the upper side of the weld layer center zone, where the highest residual stress values were to be expected.

*2.2. Thermo-Mechanical Simulation Model and Boundary Conditions*

2.2.1. Basic Equations for Thermo-Elastic-Plastic Structural Analysis

A thermomechanical FE model was developed to determine the deformations and residual stress at tool contours that are caused by 3DPMD. Therefore, FE simulations were carried out as time-dependent variables with a non-linear process behavior for temperature field calculation and, based on this, thermal-elastic-plastic structural-mechanical analysis for the calculation of deformations and residual stresses. As a first step, the implemented heat source model was used to model and numerically calculate the time-dependent 3D temperature distribution in the deposition layer contours and their environment. Subsequently, a coupled thermo-elastic-plastic structural analysis for the welded layer contours on the demonstrator component tool was realized. The transient temperature field distribution and the resulting component deformations (i.e., residual stresses) were determined and analyzed; further, significant influencing variables on their formation were determined to minimize them. Then, material, constructive, and technological measures were tested and evaluated.

The thermo-elastic-plastic simulation analysis can be carried out using the defined equations of condition, whereby the linkage of external loads and the resulting displacements takes place via so-called stiffness systems [21,22]. The basic principle for the calculation of the 3DPMD deformations and residual stresses is dependent on the temperature field variation and the phase transformation according to the mechanical material properties such as strain and plasticity. To formulate the structural-mechanical problem completely, involvement of the initial mechanical and boundary conditions is necessary to the basic equations. To simulate the heat effect during 3DPMD, the total heat flow introduced into the component is to be determined. This can be determined according to the 3DPMD parameters used, considering the thermal efficiency and the radial distribution of the heat flux density of the plasma arc. The temperature field distribution is described by the transient and nonlinear heat balance equations of the Fourier–Kirchhoff heat conduction, as shown in Equation (1) [22,23]:

$$c\rho v_{\text{w}}\frac{\partial T}{\partial t} = \nabla(\lambda\nabla T) - \nabla(c\rho\,\vec{v}\,T) + Q_{\text{T}} \tag{1}$$

If the molten pool convection is neglected for a moving heat source with a constant welding speed ($v_w$) and a torch power ($Q_T$), Equation (2) is expressed in a coordinate system moving with the source:

$$c\rho v_w \frac{\partial T}{\partial t} = \nabla(\lambda\nabla T) + Q_T \qquad (2)$$

Base material and filler materials are considered homogeneous and isotropic, but the thermophysical parameters thermal conductivity ($\lambda$), specific heat capacity ($c$) and density ($\rho$) are function of the temperature. 3DPMD assumes an ellipsoid-moving heat source on a localized distribution spot with a normally distributed heat flux density in the form of a Gaussian function (see Figure 2). For the modeling of the boundary conditions, temperature-dependent heat losses in the form of heat radiation and heat transfer are used on all sides with the heat transfer coefficient due to free convection ($\alpha_k$), the emission coefficient ($\varepsilon$) and the radiation coefficient ($C_0$), whereby the radiation is significant only on the upper side (on the underside of the component $\varepsilon = 0$, $T$—component temperature and $T_0$—ambient temperature).

$$-\lambda\frac{\partial T}{\partial n} = \alpha_k(T - T_0) + \varepsilon C_0(T^4 - T_0^4) \qquad (3)$$

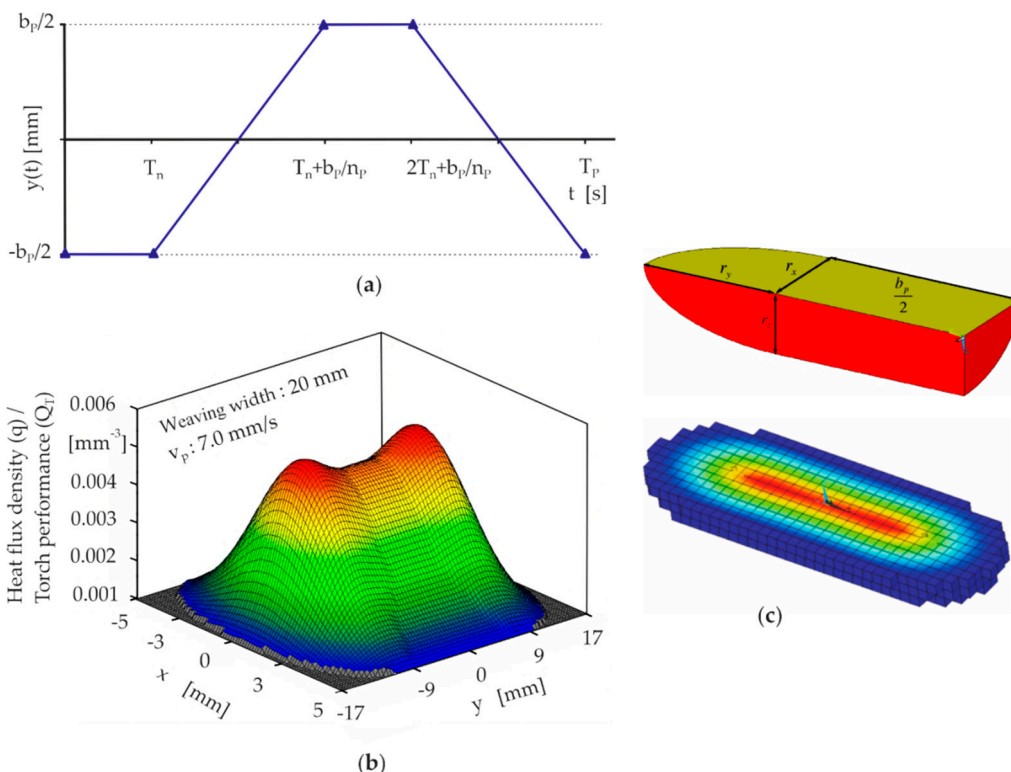

**Figure 2.** Heat source model—3DPMD with an ellipsoid PMD heat source: (**a**) finite element (FE)—heat source model; (**b**) Weaving thermal deposition and (**c**) deposited material volume produced by the heat source.

With the local heating of the component, an uneven thermal expansion formed, whereby the colder environment hinders the expansion of the warm areas. This leads to component stress formation. The stresses can reach the yield point, which decreases with increasing temperature, resulting in plastic deformations. After cooling down, residual stresses and deformations remain. The total elongation is composed of elastic, plastic, and transformation-induced and thermal elongation:

$$\varepsilon_{tot} = \varepsilon_{el} + \varepsilon_{pl} + \varepsilon_c + \varepsilon_{th} \qquad (4)$$

With many materials, a linear-elastic material behavior only applies in a limited initial range of the load. The deformation of the material caused by a load is reversible in this case. This means no deformations remain after the material has been relieved. The elastic elongation is based on Hooke's law. The elastic modulus and the transverse contraction coefficient, which are temperature-dependent, are necessary for the calculation of the elastic strain components. As a result of high loads, the stress-strain behavior becomes non-linear. If the stress exceeds the yield stress, the material is plastically deformed. This plastic deformation reduces the stresses and appears after the relief as a permanent deformation in the workpiece. In order to determine the plastic strain, it is necessary to combine the flow condition, the flow law, and the hardening law to determine the load limit of the material used.

### 2.2.2. Heat Source Model for the 3DPMD Process

In the structural simulation of 3DPMD processes, the heat input introduced into the component by the torch is described using the developed heat source model. This determines the distribution of the source density around the weld. An ellipsoid source model, according to Goldak [23,24], was used to represent the 3DPMD heat source during the process. A Cartesian coordinate system with local coordinates $x$, $y$, $z$ was introduced in the welding point, whose $x$ axis points in the move direction and whose z axis points in the normal direction of the torch. It was assumed that the heat source flow density ($q$) within a semi-ellipsoid ($E$) was associated with:

$$E = \left\{ (x, y, z) \left| \frac{x^2}{r_x^2} + \frac{y^2}{r_y^2} + \frac{z^2}{r_z^2} \right| \leq 1,\ z \geq 0 \right\} \tag{5}$$

is normally distributed, i.e.,

$$q(x, y, z) = q_0 \cdot \exp[-C_1 \cdot x^2 - C_2 \cdot y^2 - C_3 \cdot z^2],\ \text{for } (x, y, z) \in E \tag{6}$$

where $C_1$, $C_2$, $C_3$ are parameters. The still unknown maximum source strength ($q_0 = 0, 0, 0$) is determined so that the power ($Q$) transmitted by the 3DPMD heat source corresponds to the torch power ($Q_T$). i.e.,:

$$Q = \int_E q(x, y, z) \mathrm{d}V \overset{!}{=} Q_T \tag{7}$$

The power input by the torch ($Q_T$) is determined by the welding current ($I$), welding voltage ($U$) and torch efficiency ($\eta_T$):

$$Q_T = \eta_T U \cdot I \tag{8}$$

From this, it could be show the heat source that introduced by Goldak transmits only about 89% of the torch power ($Q_T$). In these investigations ($q_0$) was defined in such a way that Equation (7) is exactly fulfilled:

$$q_0 = \frac{6\sqrt{3Q_T}}{C r_x r_y r_z \sqrt{\pi^3}} \tag{9}$$

The source intensity-heat flux density distribution is, therefore, described by inserting the individual equations:

$$q(x, y, z) = \begin{cases} \frac{6\sqrt{3Q_T}}{C r_x r_y r_z \sqrt{\pi^3}} \exp\left[ -\frac{3x^2}{r_x^2} - \frac{3y^2}{r_y^2} - \frac{3z^2}{r_z^2} \right], & \text{for } (x, y, z) \in E \\ 0, & \text{otherwise} \end{cases} \tag{10}$$

Four parameters are, therefore, required for the ellipsoid model:

- the torch power ($Q_T$),
- the half-axes of the half-ellipsoid $r_x$, $r_y$, and $r_z$.

In addition to the advancing movement along the layer axis, oscillating movement perpendicular to this was carried out for a ready coating layer, see Figure 2a. The oscillating movement ideally consists of four sections:

- Torch still-situation at a reversal point during a dwell time ($T_d$),
- Movement of the torch at a constant speed ($v_p$) to the second reversal point,
- Still-situation of the torch at the second reversal point during the dwell time ($T_d$),
- Movement of the torch at a constant speed ($v_p$) to the first reversal point.

The distance between the two reversal points is the so-called weaving width ($b_p$). The period duration ($T_p$) of the weaving can thus be calculated:

$$T_p = 2T_d + 2\frac{b_p}{v_p} \tag{11}$$

In weaving thermal deposition, in addition to the forward movement at a constant speed ($v_w$), the torch head performs a fast weaving movement perpendicular to the forward movement ($v_w$) at the weaving speed ($v_p$) over the weaving width ($b_p$), so that applies:

$$\begin{pmatrix} x \\ y \end{pmatrix} = \begin{pmatrix} x_0 + v_s \cdot t \\ b_p \sin(\frac{\pi \cdot v_p}{2b_p} \cdot t) \end{pmatrix} \tag{12}$$

It is possible to simulate such a weaving motion with a moving ellipsoid source according to Equation (10), but the temporal discretization of the transient heat conduction problem must be very fine. This increases the calculation time, which can be very long, especially with a three-dimensional simulation. Therefore, a substitute source $q(x, y, z)$ was modeled, which is a time average of an ellipsoid heat source over a period of weaving motion:

$$q(x,y,z) = \frac{1}{T_p} \int_0^{T_p} (x, y_p(t) - y, z)\mathrm{d}t \tag{13}$$

By measuring the weaving travel over time using displacement transducers, the heat source distribution can be precisely defined over the layer width. Here the variation of the weaving width, speed and dwell time in turning points can be determined, see Figure 2. In weaving thermal deposition, the high frequency compared to the advancing speed results in an almost rectangular, wide layer. As heat input, higher heat flow is caused at the reversal points of the torch.

2.2.3. Discretization of Component Geometry and Definition of Temperature-Dependent Material Data

Component modeling was carried out with regard to the geometry, the temperature-dependent material data, and the associated initial and boundary conditions. FE calculations were carried out on the CAD model according to the specified component dimensions (see Figure 3a). Non-linear transient temperature and stress field analysis was realized with suitable element types by using the software package ANSYS® (different versions, ANSYS CADFEM, Grafing/Münich, Germany). The model geometry was discretized with 20 node elements "Solid 90", which are suitable for a 3D case. The FE mesh consists of a higher number of elements and finer mesh density in the coating layers and in the heat-affected zone (HAZ) to determine the large temperature and stress gradients to be expected, as shown in Figure 3b. The FE model was used to investigate the influence of 3DPMD process parameters, the number of weld layers (hard and layer contours), and constructive-technological measures such as component preheating and fixed clamping on component deformation and internal stresses.

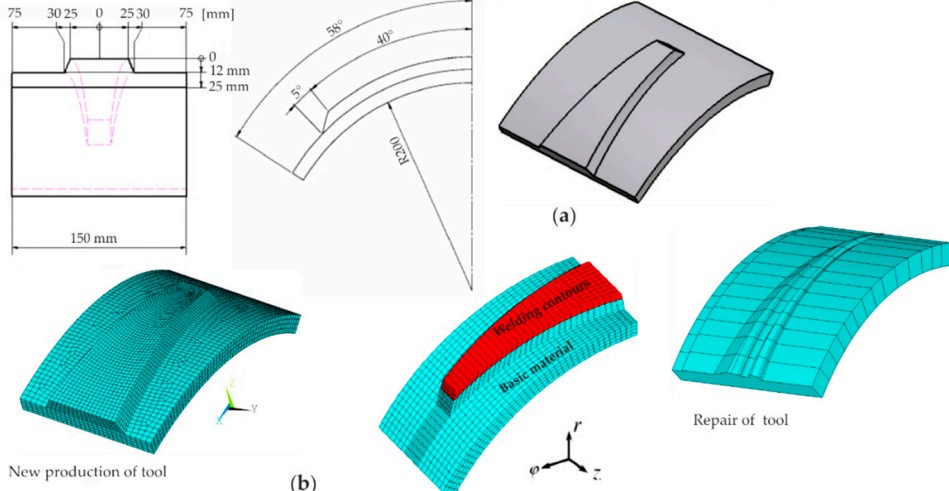

**Figure 3.** Geometry of the tool: (**a**) CAD (computer aided design) model, (**b**) FE model.

The necessary temperature-dependent thermomechanical material properties of the used materials, such as base material (i.e., hot work tool steel X40CrMoV5-1 "1.2344") and filler materials (i.e., iron-based alloy [PS Fe-hard D] as a hard layer and nickel-based alloy (Ni 625) as gradient layer) were taken from the literature sources [25–28] and from material manufacturers. These data were used as temperature-dependent functions for the FE calculations. The coefficient of thermal expansion is among the determining variables for the calculation of thermomechanical stresses; therefore, the values for the heating process from room temperature to melting temperature and the cooling process of the materials used were determined and integrated into the model. The yield strength and ultimate tensile strength characteristics of the materials as a function of temperature were also integrated into the model. In addition to the thermal boundary conditions, the temperature distribution in the component depends strongly on the heat transfer coefficient ($\alpha_k$) and the emissivity ($\varepsilon$), which were also considered in the calculations. These were recorded and described using the equations described in Section 2.2.1. As a mechanical boundary condition, the rigid body movements were excluded. The simulation calculations were, therefore, carried out under the following:

- Preheating of the basic body to $T = 450\,°C$;
- Active thermal deposition process and short cooling time to read interpass temperature;
- Active thermal deposition process up to the thermal deposition of the entire contour surfaces; and
- Long cooling time up to complete component cooling.

## 3. Results and Discussion

### 3.1. Investigation Results of the Plasma Deposition-Welded Layers

The metallographic examination results shown in Figure 4 of the iron-base deposition layers reveal the distribution of the individual phases of the structural constituents of these alloys. The coating variants of the iron-based alloys are flawless. The hard phases of the martensitic carbide microstructure and the bonding zone between the base material and the filler material are clearly visible. In addition, fine-grain formation in the transition area between individual weld layers and base and filler material can be detected in all deposited welded specimens due to the tough martensite formation with a homogeneous and fine-grained distribution, which can have a positive effect on the wear properties of these layers. The chromium carbide precipitates in the layer area are fine and small and well distributed in the matrix, which indicates a high hardness due to the high proportion of tungsten. Figure 4b also shows the micrographs of the powder composition 40% PS Fe-hard D + 60% EuTroloy 16604 with four coating layers. The material transition from the additive to the base material is

abrupt and straightforward and without bonding defects. A fine martensitic microstructure can be observed in the top layer, which mainly has light coloring. The dark and, therefore, hard part is small. In addition, the structure has coarse grains, and the precipitated carbides at the grain boundaries are large. The chromium carbide content increases with increasing proximity to the outer edge. In a comparison with the premixed PS Fe-hard D variants with different percentages with the EuTroloy 16604 powder, all the property values for the welded powder variant with 40% PS Fe-hard D + 60% EuTroloy 16604 show optimal results.

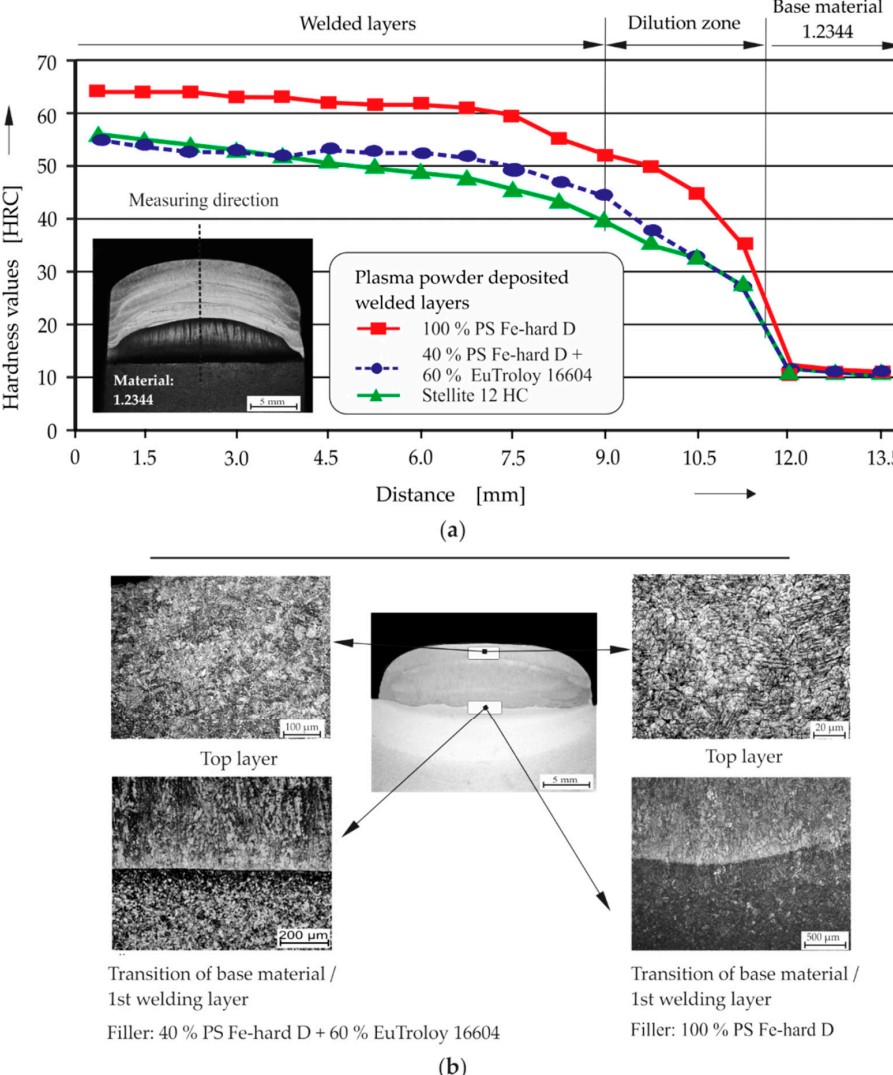

**Figure 4.** Weld metal structure and hardness curve of the plasma deposition-welded iron-base alloy: (**a**) measured hardness profiles over the welded layers depth and (**b**) microstructure after PMD in different areas.

Figure 4a shows the hardness of the deposited layers for the iron- and cobalt-based alloys. From the hardness measurements, it can be seen that the deposited layers with the filler material 100% PS Fe-hard D have the highest hardness values of 65 HRC. Furthermore, the diagram shows that the deposited samples with mixed powder 40% PS Fe-hard D + 60% EuTroloy 16604 have a maximum hardness value of 57 HRC, while the deposit-welded layers of Stellite 12 HC have a maximum hardness value of 55 HRC. This agrees with the aim of this alloy composition regarding the required layer hardness.

The examination results of the three-body abrasive sliding wear shown in Figure 5 indicate a clear trend. The hardness is not directly proportional to the wear resistance. The cobalt-based alloys

Stellite 12 HC and PS Fe-hard D should be highlighted, whereby the filler material Stellite 12 HC was used as reference material in the wear tests. The weight loss of this material is lower than that for the iron-based alloy 40% Fe-hard D + 60% EuTroloy 16604. The wear resistance of Stellite 12 HC and PS Fe-hard D was three times higher. Thus, the iron-based alloy PS Fe-hard D shows, in addition to Stellite 12 HC, a significantly higher wear resistance to abrasion under thermal conditions as soon as a high operating temperature prevails. The austenitic structure of the cobalt-based alloy Stellite 12 HC is ductile and ensures high resistance to abrasion under pressure load. A martensitic structure, on the other hand, is hard and has high resistance to abrasion without pressure and temperature stress. The influence of cobalt is also evident in the powder mixture 40% Fe-hard D + 60% EuTroloy 16604 with regard to its good wear and high hardness at high temperature. These PDM layers show a hardness of 57 HRC as well as good wear resistance, as shown in Figure 5.

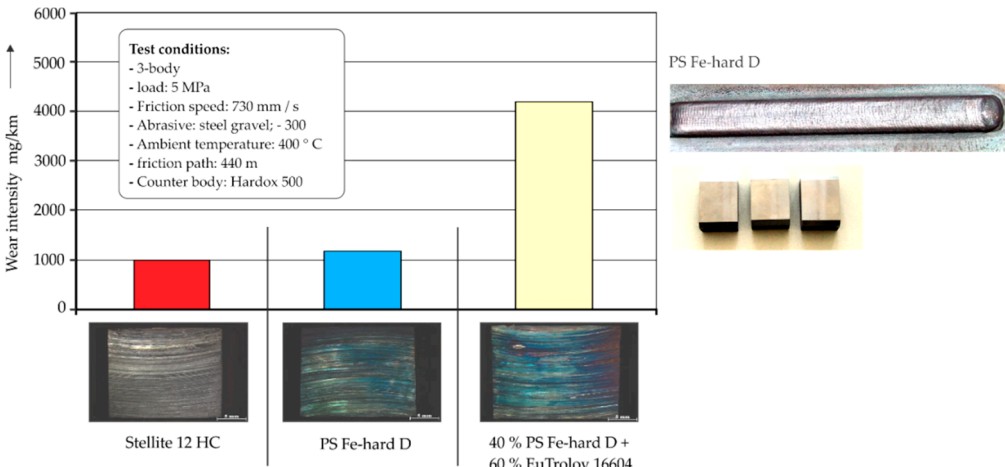

**Figure 5.** Results of the 3-body abrasive sliding wear test on the PMD layers.

### 3.2. Simulation Results

### 3.2.1. Temperature Field Distribution, Deformations and Residual Stresses

Figure 6a–c show the calculated temperature distributions on additive PDM tool structures and the temperature field distribution during component cooling at a defined cooling time. The peak temperature is more than the melting temperature of the steel, which is quite realistic. The isotherms of the temperature fields begin from a certain number of layers as linear surfaces. Their intensity depends on the distance across the deposited layer. The isotherms extend parallel to the layer axis as linear surfaces. By contrast to the first deposited layer(s), a moving heat source, the associated temperature field is quasi-stationary, i.e., the local temperatures are independent of time in a co-rotating coordinate system. Application of the first welding layers prevail on the component a transient temperature field distribution due to the good heat dissipation of the component. Upon reaching a certain number of layers, the heat removal condition changes greatly (limited heat dissipation in the body and high radiation and convection in the weld metal area). This depends on the dimensions of the components, the heat input, the type of coating material, and the interpass temperature and thus influences the resulting mechanical properties of the welding contours.

This means that after a specific transient temperature depending on the number of layers the base body strives for a limit state in which the temperature is balanced over the entire component and the temperature gradient in the area of the heat source (melt pool) increases. At the beginning of the first thermal deposited layer, the internal heat dissipation predominates with 3D or 2D heat modeling, depending on the component dimensions. The heat dissipation is severely restricted by the increase in the deposited volume. The proportions of heat convection and radiation in the heat balance increase with the number of layers and the deposited volume. In the lower part of the applying volume and

in the base body, the temperatures usually decrease with an increase in the number of layers until a certain temperature value is reached.

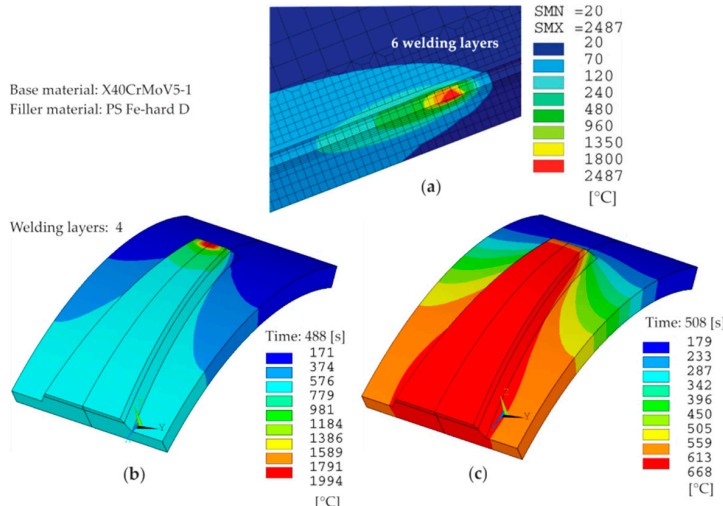

**Figure 6.** Temperature distribution for additive PMD tool: (**a**) linear thermal deposition; (**b**) weaving thermal deposition and (**c**) component cooling.

In Figure 7a,b, the residual stress condition of the additive deposit welded tool structure surfaces can be seen after four layers and complete cooling of the component to room temperature. According to the stress state, tensile stresses develop in the deposition layer area that have their maximum value in the middle of the layer $\sigma$ = 734–829 MPa. In the adjacent areas, tensile stresses also occur, which decrease with more distance across the center. In the area of the base material (base body), smaller tensile stresses develop, which reach a maximum value of $\sigma$ = 100 MPa and drop to zero at the edge of the component. In deeper areas, the residual stresses in the deposited layer and base material areas are reduced. Figure 7c shows the distribution of the FE calculated deformation values perpendicular to the component plane after complete cooling of the component to room temperature. It can be seen that a component contraction and deflection occurred at the component due to the plastic deformation material areas and the shrinkage of the deposited welded layers.

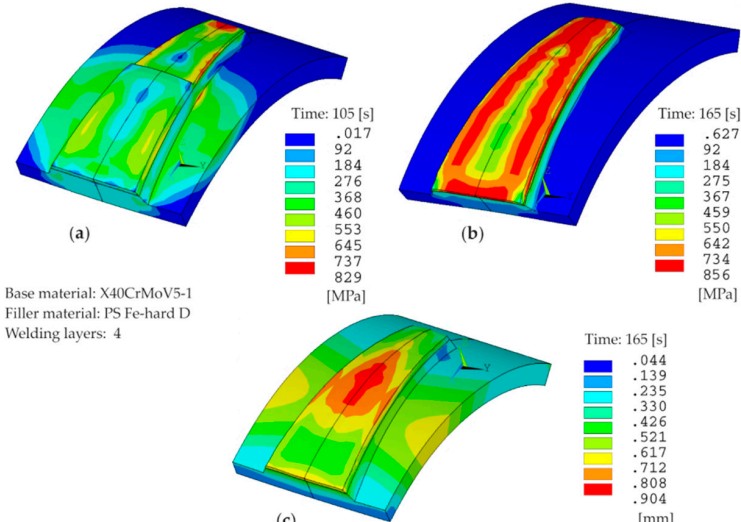

**Figure 7.** Simulation results: (**a**) comparative stress according to Van Mises during the thermal deposition second hard layer; (**b**) comparative stress according to Von Mises after complete component cooling; and (**c**) deformation perpendicular to the component plane after complete component cooling.

### 3.2.2. Comparison of Finite Element (FE) Results with Experiments

In the first step, the thermal partial model was verified regarding calculated temperatures. The temperatures were measured by attached thermocouples at the deposit layer edge and the base material single of and multilayer thermal deposition (the same locations for FE calculations and for applied measurements by thermocouples, see Figure 8a). The calculated and measured temperatures were analyzed and compared (see Figure 8a). This showed good agreement between the calculated and measured values. The difference between calculated and measured values near to the source area was 12%. The additive deposition-welded layer geometries were well represented mathematically, in comparison with Figure 8b.

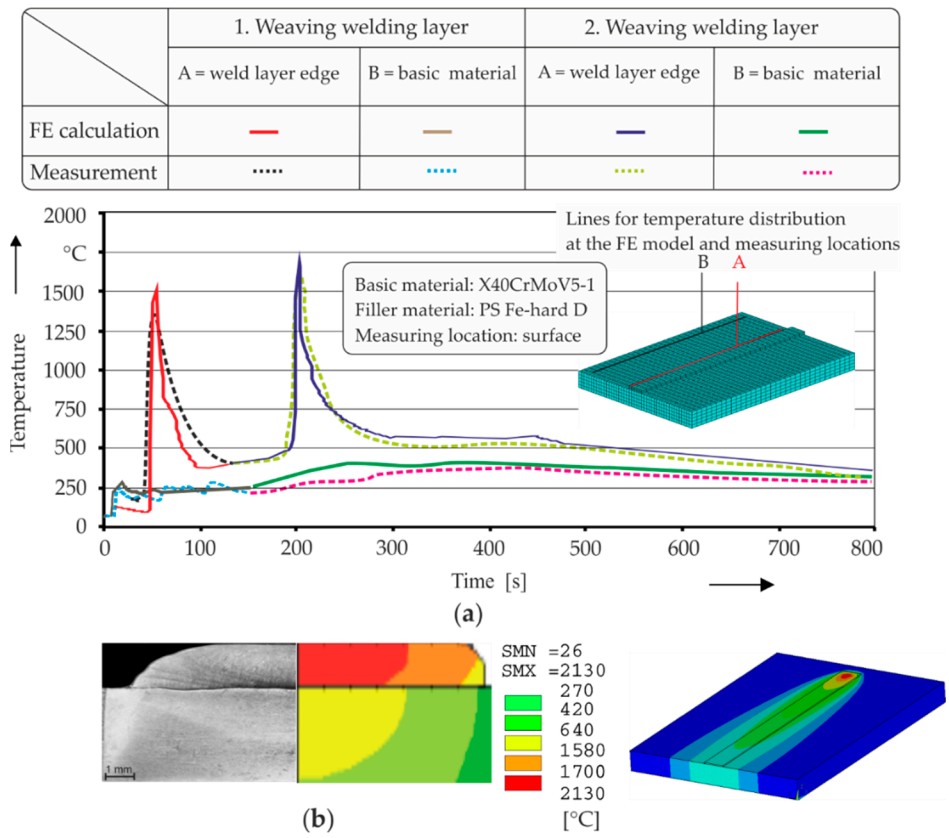

**Figure 8.** Comparison between calculated and measured values: (**a**) comparison—temperature and (**b**) comparison—layer geometry by weaving thermal deposition.

Furthermore, the mechanical model was validated regarding the calculated thermal deposition deformations and residual stresses. The calculated values of the dimensional and geometrical deviations for four deposited layers were compared with the measured values using a measuring coordinate system. Measurement of the residual stresses was performed at the last layer (fourth layer) with a depth of 1.0 mm. The calculated values showed good agreement with the measured values. The difference was 10%. Verification of the FE model regarding the residual stress behavior was carried out depending on the measured stress values using the hole-drilling method [20] in the middle of the coating layer zones (see Figure 9). The calculated longitudinal and transverse residual stress values showed good agreement with the measured values. The difference was ≤15%. Thus, the FE thermomechanical model was further used for determination of the influence of the material-technological measures to minimize the deformations and residual stresses on the additively deposited component.

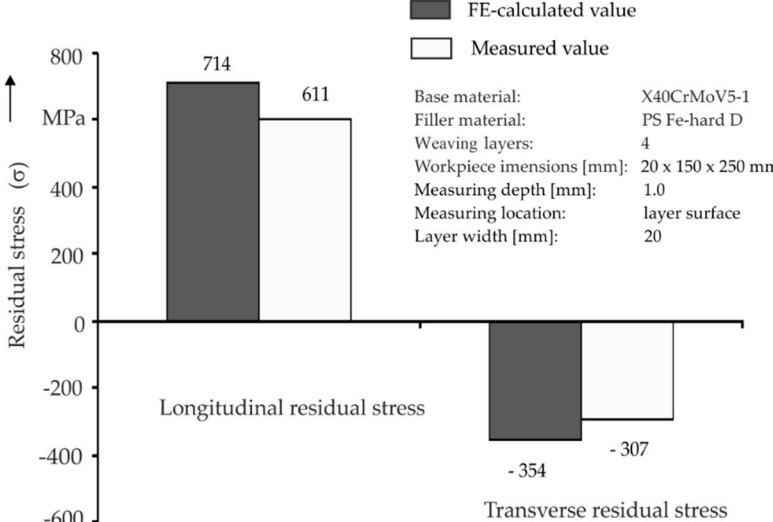

**Figure 9.** Comparison between calculated and measured stresses using the hole-drilling method.

### 3.2.3. Material Technological Means to Minimize Deformation/Residual Stresses on Additive Plasma Deposition-Welded Component Structures

After the thermomechanical simulation model was verified, the model was used to minimize/compensate the deformations and residual stresses in the additively manufactured component. Preheating, fixed clamping, and 3DPMD of ductile layer materials were involved in the investigations. The influences of these measures on the deformation and residual stress scenario during 3DPMD of the tool contour surfaces were determined and evaluated. A preheating temperature of $T = 450$ °C was chosen according to the used base material and then modeled within the boundary conditions. Preheating ensures that the component is uniformly heated, which means that the temperature gradient of the quasi-stationary thermal deposition temperature field between the base material and the deposit layers is no longer large. This results in low strain and compression and, consequently, lower deformation/residual stresses.

The calculated values with the preheating temperature showed a deformation and residual stress reduction of approximately 20% compared with the calculated values without component preheating (see Figure 10). During the 3DPMD process, however, higher stresses arise on the firmly clamped base body due to the shrinking cross section, which can lead to component cracks. Stress due to constrained angular shrinkage can be superimposed. This results in higher compressive stresses in the transverse direction in the deposited layers, which are to be regarded as the sum of both reaction and restraint stresses. The thermal deposition deformation is reduced, but the residual stresses increase as the free shrinkage is restricted. Figure 10 shows the calculated stress values on the fixed component after it has completely cooled down. In the fixed component, the stress value was 131 MPa higher than in the non-clamped component (see Figure 9). However, the fixed clamped component shows low deformation (max. deflection: 0.218 mm) in comparison with the non-fixed clamped component (max. deflection: 0.904 mm). The avoidance of cracks due to high stresses can be achieved by 3DPMD of the component contour with ductile layer materials in connection with a heat effect optimized for the used component and the filler materials. Applying a ductile layer gives better stress characteristics on the hard layer/functional surface than without applying ductile layer materials. Due to the different thermomechanical properties of the material, large component deformation and residual stress take place. It was, therefore, necessary to use a sufficiently ductile and high-temperature-strength material for the 3DPMD of the component contour surfaces. Figure 10 shows the influence of the applied ductile layer structures on the residual stresses during thermal deposition of the component contour surfaces. Based on the maximum stress values at the layer surface, a 15–30% reduction in the stress peaks and deformation values was achieved with a ductile deposition layer of Ni 625 alloys and three hard coating layers of PS Fe-hard alloys D compared with the four hard coatings with PS Fe-hard

alloys D without a ductile coating layer. This shows the advantage of additive 3DPMD of component contour surfaces with ductile layers to reduce residual stresses under an optimal thermal regime.

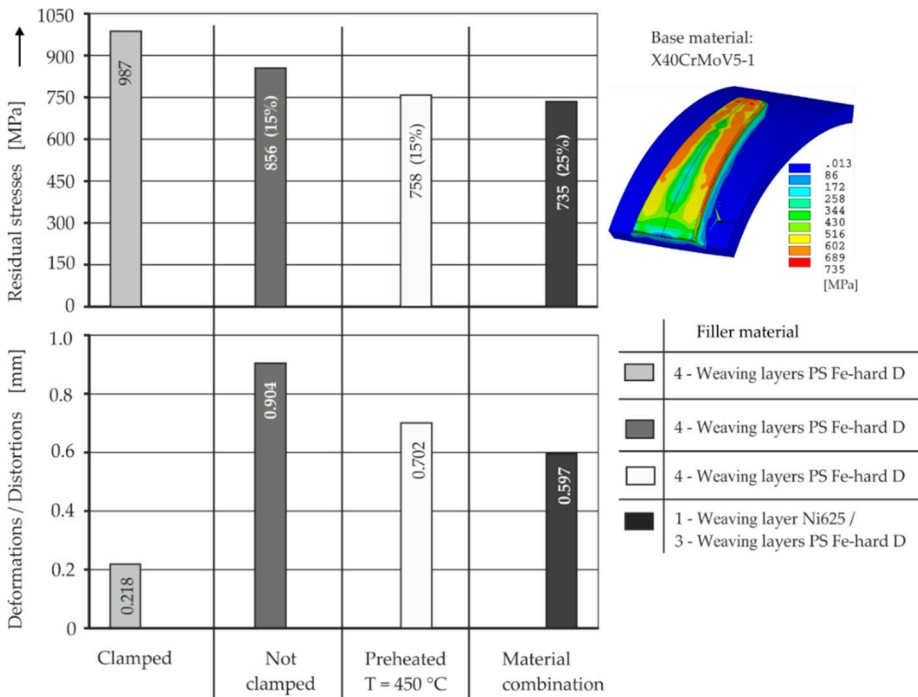

**Figure 10.** Using of material-technological means for minimization of residual stresses/deformations at additive plasma deposition-welded tool structures.

## 3.3. Additive 3DPMD of Structural Component Surfaces

A 3DPMD technology for the production of component contour surfaces of the tool geometries was developed to prove the practical implementation of the obtained test results. The functional surfaces of the first segment of a cross-rolling tool with nickel-based alloys and even mixed iron-base alloys were 3DPMD manufactured. The tool geometries were additive manufactured on a base body made of a high-alloy 1.2344 tool steel regarding the 3D-CAD model as well as the defined filler materials in multilayer technology (see Figure 11). This and a change in the thermal deposition direction during 3DPMD of the subsequent weld layer led to the minimization of component distortion and residual stresses. Due to the limited weldability of the base body material used (1.2344, see Table 1), the component was preheated to a temperature of 450 °C to avoid possible cracks. The investigation results regarding the optimal layer additive system and its material alloys and the wear investigations were further used in the manufacture of the tool structure surfaces. The applied deposition materials were the nickel-based alloy Ni 625 for the 3DPMD of tool wear surfaces and the iron-based alloy PS Fe-hard D for the production of wear-resistant tool functional layers. To protect the surfaces of the basic tool body from the effects of wear during the forming process, PMD layers were applied. After completion of the additive 3DPMD of the tool contour surfaces, surface inspection was carried out using dye penetrant testing. This test showed no cracks or layer defects on the component surfaces (see Figure 11). The results obtained come from using the many strategies mentioned above, e.g., preheating and using ductile materials (Ni 625) in the first layers.

Basic material: X40CrMnV5-1; Filler materials: Ni 625 und PS Fe-hard D

**Figure 11.** Additive 3DPMD technology: (**a**) build-up welded tool contours—1. segment of cross rolling tool and (**b**) welded tool contours after dye penetration test.

## 4. Summary—Conclusions

- Using sophisticated additive 3DPMD, complex component geometries with predefined thermomechanical properties can be produced from large weld metal volumes. It possible to produce complex geometries with their shapes, functions and thermomechanical properties.
- 3DPMD thus enables the layer-by-layer production of metallic components based on a virtual CAD component model. By mixing several powders in an arc, the local properties of the deposited layers can be adapted locally to the defined service loads.
- With the layer-by-layer construction system that was developed on the basis of the selected and self-mixed thermal deposited powder alloys made of iron-based alloys, high wear resistance and high hardness at high temperature under the defined thermal conditions as well as crack-free component contour surfaces were additively plasma metal manufactured.
- A thermo-mechanical simulation model was successfully developed, validated and further used for the predetermination/minimization of deformation and residual stresses on 3D plasma deposition welded structures for complex component geometries.
- Depending on the alloys used, complex shrinkage and transformation stresses occur in the area of the thermal deposited contours.
- The difference between the FE calculations and the measurements is approximately 15%, and this shows the practical application potential of the simulation model.
- A minimization of the deformation/residual stresses on plasma deposition-welded component structures was demonstrated using preheating, fixed clamping and PMD with a ductile layer material.
- The knowledge gained was implemented in practice by producing complex, highly stress-resistant component geometries with defined layer properties–tool contours.

**Author Contributions:** K.A. designed and realized the experiments and developed the numerical simulation of the 3DPMD process as well as writing the paper; P.M. was the project coordinator and supervisor.

**Funding:** The presented experimental and theoretical investigations were supported by the German Research Foundation (DFG). We would like to express our gratitude for this support.

**Acknowledgments:** The authors thank the colleagues at the Chair of Welding Engineering, Chemnitz University of Technology for their unlimited support for this work.

**Conflicts of Interest:** The authors declare no conflict of interest.

## Nomenclature

| Symbols | Unit | Definition |
| --- | --- | --- |
| $b_p$ | m | Weaving width |
| $C$ | - | Constant |
| $c$ | $J \cdot kg^{-1} \cdot K^{-1}$ | Specific heat capacity |
| $C_0$ | $1.380651 \times 10^{-23}$ $J \cdot K^{-1}$ | Stefan–Boltzmann constant |
| $E$ | - | Half ellipsoid |
| FE model | - | Finite element model |
| $I$ | A | Welding current |
| $Q$ | W | Total performance |
| $Q_T$ | W | Energy effect–torch power |
| $q$ | $J \cdot m^{-2} \cdot s^{-1}$ | Source intensity–heat flux density |
| $r_x$, $r_y$ and $r_z$ | m | Half axis of the half ellipsoid |
| $T$ | K | Temperature |
| $t$ | s | Time |
| $T_p$ | $m \cdot s$ | Period duration |
| $T_d$ | s | Dwell time |
| $U$ | V | Welding voltage |
| $v_w$ | $m \cdot s^{-1}$ | Welding speed |
| $v_p$ | $m \cdot s^{-1}$ | Weaving speed |
| $\rho$ | $kg \cdot m^{-3}$ | Density |
| $\lambda$ | $W \cdot m^{-1} \cdot K^{-1}$ | Thermal conductivity |
| $\alpha_k$ | $W \cdot m^{-2} \cdot K^{-1}$ | Heat transfer coefficient |
| $\varepsilon$ | $W \cdot m^{-3} \cdot sr$ | Emission coefficient |
| $\varepsilon_{el}$ | - | Elastic strain |
| $\varepsilon_{tot}$ | - | Total strain |
| $\varepsilon_{pl}$ | - | Plastic strain |
| $\varepsilon_{th}$ | - | Thermal strain |
| $\varepsilon_c$ | - | Conversion-induced strain |
| $\eta_T$ | % | Torch effectivity |
| $\sigma$ | MPa | Surface stress |
| $\nabla$ | - | Nabla-Operator |
| 3DPMD | - | Three dimensional plasma metal deposition |

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
