# Peer review of "Additive Manufacturing of Complex Components through 3D Plasma Metal Deposition—A Simulative Approach"

_metals, doi:10.3390/met9050574_

Round 1
Reviewer 1 Report
Very interesting and relevant work. The scope of the work is well-detailed and the methodologies are most often well explained besides some, which are mentioned in the comments below.
This work is certainly valuable for any researchers and industry working in the field of metal AM and simulation of manufacturing processes.
It is recommended to incorporate/address the suggested comments below and to expand the introduction. It is not clear how this work stands out in originality of other work done in this field. Some more context and references of other work in this field needs to be given.
Line 33 : 3D Plasma Metal Deposition (PMD) à make sure that notations/abbreviations are consistent throughout text
Line 35: No post processes needed after 3DAMD?
Line 41-42-43 : revise sentence as it is not clear exactly what is stated here.
Line 45: max. layer height is mentioned, what about the min. layer height per deposition pass? Useful information to classify this 3DPMD technology.
Line 65: Finite Element (FE)
Line 67: precise for which application fields the tool was designed (i.e. which requirements on its surface are key?)
Line 92-104: “2.1.2 Process parameters…..system” à which methodology was use to find the best parameters to use for the 3DPMD process? This information is lacking.
Line 108-123: It is suggested to put all tested and required parameters in a table to easily compare how well the process performs.
Line 240: Heat Affected Zone (HAZ)
Line 255: how are the flow and hardening behavior of the materials obtained? Existing data or experiments carried out?
Line 349-351: an overview (image) of the setup with thermocouples attached at the layer edges could be given to have a better idea of quality of the comparative study.
Line 428: clarify why these images show good results (i.e. how can it be observed that no cracks are present?)
Line 434: what is meant by “complex” geometries?
Author Response
· Line 33 : 3D Plasma Metal Deposition (PMD) à make sure that notations/abbreviations are consistent throughout text
All notation/abbreviations are consistent
· Line 35: No post processes needed after 3DAMD?
In some cases it need less post processing such as mechanical machining in compared with the traditional manufacturing process, for example production of complex shape with many inside channels. However, in most cases it leads to reduce the amount of the amount of the scrap materials which, leading to reduce costs. The content of line 35 is changed to according to this fact.
· Line 41-42-43 : revise sentence as it is not clear exactly what is stated here.
The sentence is cleared in the line 41-43.
· Line 45: max. layer height is mentioned, what about the min. layer height per deposition pass? Useful information to classify this 3DPMD technology.
There is no clear separate limit between the thickness of the deposited layer in the deposition welding and the additive manufacturing process. It always depending on many factors such as welding parameters, deposited materials, base material. In general, the minimum thickness of the deposition layer, needs less energy this means less welding parameters (low current, low voltage), and to insure the stability of the plasma arc a microplasma process is used. Generally with this process the minimum thickness of the deposit layer is from 1-5 mm.
· Line 65: Finite Element (FE)
Already change
· Line 67: precise for which application fields the tool was designed (i.e. which requirements on its surface are key?)
The precise tool is cross-rolling tool: It is used for rolling process with special profile
Line 92-104: “2.1.2 Process parameters…..system” à which methodology was use to find the best parameters to use for the 3DPMD process? This information is lacking.
23 Two-level method was used
· Line 108-123: It is suggested to put all tested and required parameters in a table to easily compare how well the process performs.
The experimental parameters are not listed in a table due to the main part of this work was the simulation process. The experimental work will be listed in other research work dependent more on the studying the effect of the parameters on the 3DPMD process. These experimental work will be discussing in more details.
· Line 240: Heat Affected Zone (HAZ)
Already changed
· Line 255: how are the flow and hardening behavior of the materials obtained? Existing data or experiments carried out?
There is a miss understanding due to the wrong used words. Now it is more clear.
· Line 349-351: an overview (image) of the setup with thermocouples attached at the layer edges could be given to have a better idea of quality of the comparative study.
A sentence was added to describe the positions of the thermocouples. The positions of the thermocouples are the same in calculated and measured.
· Line 428: clarify why these images show good results (i.e. how can it be observed that no cracks are present?)
The obtained results come from the using of many strategies that mentioned above such as pre-heating, using of ductile materials (Ni 625) at the first layers.
· Line 434: what is meant by “complex” geometries?
Complex geometries: all geometries that is not easy to produced using conventional methods due to their shapes, functions and thermomechanical properties.

Reviewer 2 Report
Interesting work on additive manufacturing technologies. Before acceptance some aspects need to be revised as follows:
The English needs to be revised by a native speaker. There are several mistakes in the text.
There are multiple references that are in German that most researchers in the field cannot read. Therefore, it is suggested to add some of the key review works in the area of additive manufacturing to support the introduction of the current manuscript. Refer to A review of wire arc additive manufacturing and advances in wire arc additive manufacturing of aluminium, Current Status and Perspectives on Wire and Arc Additive Manufacturing (WAAM) and Strategies and processes for high quality wire arc additive manufacturing as key examples.
In Figure 1 c a scale is missing.
Can the authors quantify the grain size variation along the height of the as-built parts based on the OM images? This could be interesting to be discussed in the manuscript.
“temperature is at the melting temperature of the steel, which is quite realistic”: The peak temperature during any fusion based AM process is always above the melting point of the material. This must be changed.
“With a moving heat source, the associated temperature field is quasi-stationary”: debatable. In the first deposited layers there is transient behaviour due to the fast cooling induced by the substrate. Only after some layer the temperature along the height is more or less the same. Refer to recent article who have addressed this issue as Wire and arc additive manufacturing of HSLA steel: Effect of Thermal Cycles on Microstructure and Mechanical Properties.
In which part of the as-built materials were the residual stresses measured? As a result of the thermal cycles it is expected that the residual stresses vary along the height of the sample.
How were the actual residual stresses determined? This is not presented in the materials and methods section.
Why were not the mechanical properties studied? Perhaps the authors have this data already and can add them to the manuscript?
Author Response
The English needs to be revised by a native speaker. There are several mistakes in the text.
The text was revised by a native speaker through external Agency
There are multiple references that are in German that most researchers in the field cannot read. Therefore, it is suggested to add some of the key review works in the area of additive manufacturing to support the introduction of the current manuscript. Refer to A review of wire arc additive manufacturing and advances in wire arc additive manufacturing of aluminium, Current Status and Perspectives on Wire and Arc Additive Manufacturing (WAAM) and Strategies and processes for high quality wire arc additive manufacturing as key examples.
The introduction was well edited. Many English references English were added. Many subjects were listed in the introduction such as WAAM process
In Figure 1 c a scale is missing.
Is already changed
Can the authors quantify the grain size variation along the height of the as-built parts based on the OM images? This could be interesting to be discussed in the manuscript.
This is not planed in the work due to the main part of this work related to simulation process. So the other evaluation processes that are not directly connected to the simulation were not performed.
“temperature is at the melting temperature of the steel, which is quite realistic”: The peak temperature during any fusion based AM process is always above the melting point of the material. This must be changed.
It is already changed: The peak temperature is more than the melting temperature of the steel, which is quite realistic.
“With a moving heat source, the associated temperature field is quasi-stationary”: debatable. In the first deposited layers there is transient behaviour due to the fast cooling induced by the substrate. Only after some layer the temperature along the height is more or less the same. Refer to recent article who have addressed this issue as Wire and arc additive manufacturing of HSLA steel: Effect of Thermal Cycles on Microstructure and Mechanical Properties.
This topic was discussed in details in 3.2.1. Temperature field distribution, deformations and residual stresses
In which part of the as-built materials were the residual stresses measured? As a result of the thermal cycles it is expected that the residual stresses vary along the height of the sample.
It is already added: The measuring of the residual stresses was performed at the last layer (4th layer) with depth of 1.0 mm.
Please see the paragraph 3.2.2. Comparison of FE results with experiments
How were the actual residual stresses determined? This is not presented in the materials and methods section.
It is already mentioned in the 2.1.3. Characterization of the Plasma Deposition-Welded Layer System section.
Why were not the mechanical properties studied? Perhaps the authors have this data already and can add them to the manuscript?
The part of the data was taken from the literature (Modellbildung und Simulation des Plasma-Schweißens zur Entwicklung innovativer Schweißbrenner – Modeling and simulation of plasma welding for the development of innovative welding torches, Habilitation, Chemnitz University of Technology – Germany 2017, 978-3-96100-007-4.) the other part was taken from the producers’ data base and there is a confidential agreement about this information

Reviewer 3 Report
This manuscript presents new results on additive manufacturing, including both experimental and modelling data. It is well organized and the topic is suited for publication in Metals. Please note that there is a German "und" in the caption of figure 8, which should be translated to English. The sections: Author contributions / Funding / Acknowledgements / Conflicts of Interest should be completed.
Author Response
Please note that there is a German "und" in the caption of figure 8, which should be translated to English. The sections: Author contributions / Funding / Acknowledgements / Conflicts of Interest should be completed.
The German word 'und' was changed.
The other information was added at the end of the text

Round 2
Reviewer 2 Report
There are still problems with the manuscript as detailed below. Please address them.
The references still need to be changed. Most of them are in German, which most people including the reviewer does not understand. For example, refs 1, 2, 5, 6, 7, 8, 16, 17, 18, 19, 20, 23, 26 and 27 are all in German. This must be changed. Refer to key review papers that potentially state what those German references are refereeing as suggested in the previous review: A review of wire arc additive manufacturing and advances in wire arc additive manufacturing of aluminium, Current Status and Perspectives on Wire and Arc Additive Manufacturing (WAAM) and Strategies and processes for high quality wire arc additive manufacturing.
Also, using PhD or MSc thesis as references is dubious…
One thing that is not clear yet: why the residual stresses are only measured in the last layer. The developed residual stresses will vary due to the thermal cycle and the last layer has a different thermal cycle then the already deposited ones.
Author Response
Thank you for reviewing our work. We took all suggestions in the account.
- The most references were changed into English references
- There are some references still in German language because of:
o The references (16,17) present the material data.
o the references (18,19) present the used measurements methods and standards.
o the references (25-27) present the simulation process as well as material data
- The introduction paragraph in general was edited and the introduction construction was changed.
- The WAAM process was discussed more in details in the introduction paragraph.
- The WAAM process for aluminum was discussed in the introduction paragraph.
One thing that is not clear yet: why the residual stresses are only measured in the last layer. The developed residual stresses will vary due to the thermal cycle and the last layer has a different thermal cycle then the already deposited ones.
The presented drill-hole method can be used to drill out blind holes up to a depth of 1.8 mm. From the performed investigations, the drilling in deeper distance more than 1.0 mm in hard layer gives no more reliable information regarding strain changing. The to be determined strain depend on the height of the residual stresses in the respective depth increment and the geometric conditions (drill diameter, distance of the strain gages to the drill hole edge, strain gage geometry, etc.). The stresses reached it’s maximum value at the surface of the deposited layer. The inter pass temperature (heat treatment process) reduces the stresses in deeper distances. As the number of layers’ increases, the stress values on the surface of the last weld layer increase rapidly due to the faster surface cooling of the last weld layer.
This information is also included in the paper.

Round 3
Reviewer 2 Report
The authors address the concerns of the reviewer.